# Engineering solutions for biological studies of flow-exposed endothelial cells on orbital shakers

Andreia Fernandes[1], Vahid Hosseini[1], Viola Vogel[1], Robert D. Lovchik [2]*

1 Department of Health Sciences and Technology, Eidgenössische Technische Hochschule Zürich, Zurich, Switzerland, 2 IBM Research Europe, Rueschlikon, Switzerland

* yrl@zurich.ibm.com

## Abstract

Shear stress is extremely important for endothelial cell (EC) function. The popularity of 6-well plates on orbital shakers to impose shear stress on ECs has increased among biologists due to their low cost and simplicity. One characteristic of such a platform is the heterogeneous flow profile within a well. While cells in the periphery are exposed to a laminar and high-velocity pulsatile flow that mimics physiological conditions, the flow in the center is disturbed and imposes low shear stress on the cells, which is characteristic of atheroprone regions. For studies where such heterogeneity is not desired, we present a simple cell-patterning technique to selectively prevent cell growth in the center of the well and facilitate the exclusive collection and analysis of cells in the periphery. This guarantees that cell phenotypes will not be influenced by secreted factors from cells exposed to other shear profiles nor that interesting results are obscured by mixing cells from different regions. We also present a multi-staining platform that compartmentalizes each well into 5 smaller independent regions: four at the periphery and one in the center. This is ideal for studies that aim to grow cells on the whole well surface, for comparison with previous work and minimal interference in the cell culture, but require screening of markers by immunostaining afterwards. It allows to compare different regions of the well, reduces antibody-related costs, and allows the exploration of multiple markers essential for high-content screening of cell response. By increasing the versatility of the 6-well plate on an orbital shaker system, we hope that these two solutions motivate biologists to pursue studies on EC mechanobiology and beyond.

## Introduction

Several studies have reported the importance of flow-driven forces, such as shear, on the function of vascular smooth muscle cells (SMCs) and endothelial cells (ECs) [1–3]. Under both *in vitro* and *in vivo* conditions, exposure of ECs to laminar pulsatile flow within the physiological range promotes quiescence and expression of anti-inflammatory, anti-coagulant and survival markers [4–6]. Furthermore, ECs are known to align in the direction of the flow, a morphological change that can be utilized as a marker for a healthy EC response to flow and

**Data Availability Statement:** All relevant data are within the paper and supporting information files.

**Funding:** This work was part of the Zurich Heart project from Hochschulmedizin Zürich and was

funded by the Zurich Heart Grant and the ETHeart Initiative (ETH Zurich).The funders had no role in study design, data collection and analysis, decision to publish, or preparation of the manuscript.

**Competing interests:** The authors have declared that no competing interests exist.

atheroprotective phenotype [7, 8]. On the other hand, disturbed flow is characteristic of atherosclerosis-susceptible regions, promoting a proliferative and pro-inflammatory profile, and resulting in random EC alignment [9–11]. To better visualize endothelial alignment, immunostaining against VE-cadherin is typically performed. This protein is not only part of the mechanoreceptor complex that senses the shear stress at the cell membrane of ECs, but is also the most abundant cadherin in adherent junctions [12–15].

To better mimic the native environment, various strategies have been developed to expose ECs to shear *in vitro*, by using parallel plate flow chambers, plate-cone systems, microfluidic devices, or orbital shakers [16, 17]. Many *in vitro* studies with ECs were performed using orbital shakers due to their availability in many labs, ease of use, and the large cell numbers that can be harvested for omics analyses, as summarized in Warboys et al. [9]. Moreover, no complicated tubing or pumps are required, and the technology is amenable to microscopical imaging and sampling of supernatants for the analysis of secreted factors. The only prerequisite is a standard cell culture 6-well plate on an orbital shaker. It allows studying multiple conditions simultaneously, which is especially advantageous for screening purposes.

Despite being a simple and high-content platform, one potential major drawback of the orbital shaker is that the fluid flow profile is heterogeneous within each well. As described in prior studies using computational simulations, the cells at the periphery are exposed to a relatively uniform (laminar) and high-velocity pulsatile flow, which closely mimics the physiological flow profile in blood vessels [18–20]. As could be expected, cells in the periphery align in the direction of the flow, while in the center of the well disturbed flow with low velocity causes the ECs to orient randomly, similar to pro-atherosclerotic regions of arteries [9]. This heterogeneous flow profile within each well can be a limitation of this platform, since it is technically challenging to differentially harvest biological material, such as RNA or protein, from the center and the periphery of the well. The analysis of such a mixed cell population with different phenotypes could mask potentially interesting biological responses, as shown in a study by Ghim et al. [21], where the phenotype was compared between ECs in the center and periphery in either full or segmented wells. In this study, TNF-α treated HUVEC segmentation increased monocyte adhesion and expression of VCAM-1 and ICAM-1 in the center of the well but had no effect at the periphery, suggesting that ECs at the periphery express mediators that reduce inflammation in the center. Here we present a simpler cell-patterning solution to grow cells only in the periphery, preventing cell growth in the center of the well with a PEG-gel coating (Figs 1 and 2).

Flow heterogeneity can be advantageous if the goal of the study is to compare the two adjacent regions within the well and the effect of different flow profiles without losing the big picture or interfering with the cell culture system. However, in this case, immunobiological studies in 6-well plates are a particular challenge, since the wells are relatively big and the large volume of reagents, such as antibodies, required for each well is costly. Smaller well sizes are typically not used for endothelial studies, since physiological shear stress values cannot be reached on an orbital shaker without increasing other parameters such as volume, height and rotation speed [9]. One solution would be to compartmentalize the tissue after the experiment on the shaker to reduce the area used for staining. However, previous testing showed that conventional masking methods tend to fail on such surfaces, since the tissue surface is hydrophilic and fragile. Furthermore, conventional plastic-bottom wells require long working distance microscope objectives due to their thickness, which hinders high-resolution imaging. To overcome both challenges, we used thinner glass-bottom 6-well plates and developed a 3D-printed multi-staining platform that allows the formation of five sealed compartments within one well and performing up to 16 parallel stainings (Figs 1, 4 and 5).

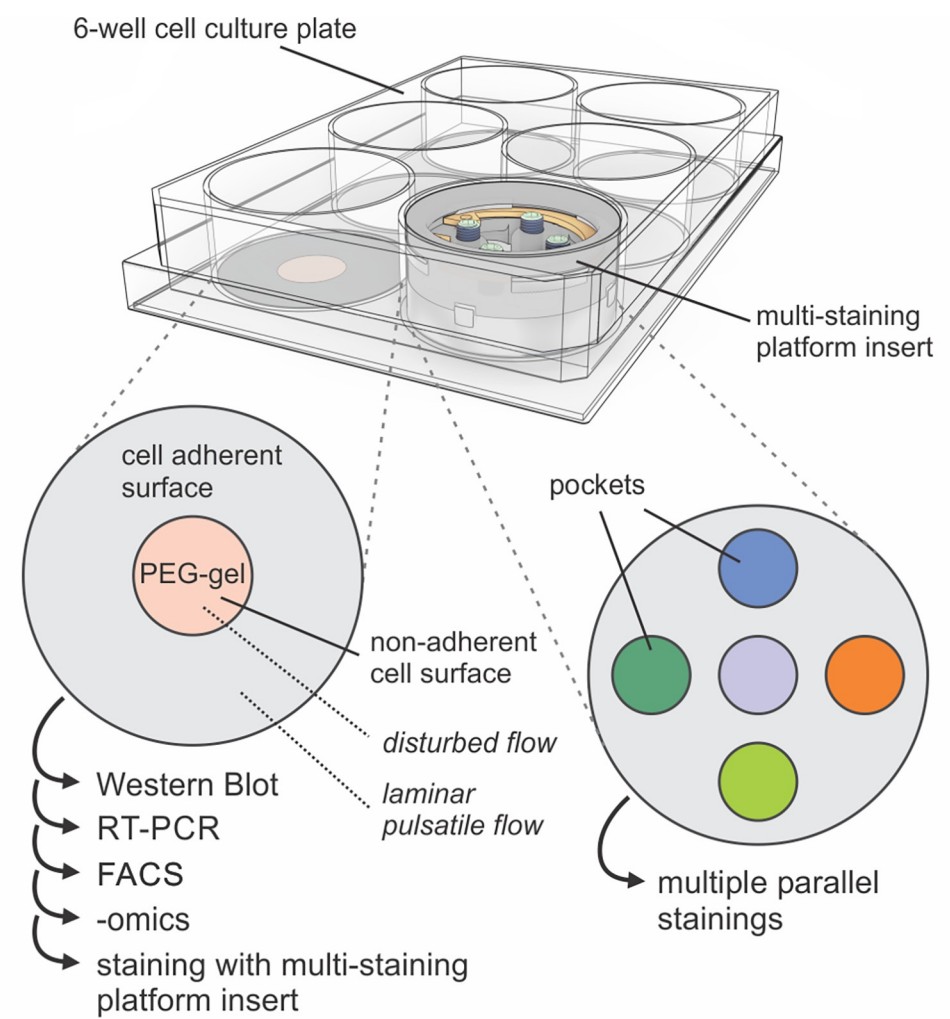

**Fig 1. Overview of a 6-well plate and the proposed solutions.** Left: Non-adherent PEG coated surface in the central region of the well, where flow is disturbed. This solution allows to selectively grow cells in the periphery where flow is physiological (laminar pulsatile). After cell culture, a more homogeneous sample can be harvested for various biochemical assays (Western blot, RT-PCR, FACs, -omics) or stained using our second solution, shown on the right. Right: Multi-staining platform insert, which can be applied after sample fixation. Five individual pockets, one in the center and four in the periphery, allow up to 16 parallel stainings, considering four stainings per pocket and a comparison between a peripheral pocket and the one in the center.

By providing detailed simple engineering instructions, we hope that some significant drawbacks of orbital shaker platforms can be overcome, while preserving their advantages, and can thus enrich the toolset of vascular biologists.

## Methods

The protocols described in this peer-reviewed article are published on protocols.io (dx.doi.org/10.17504/protocols.io.b2bwqape) and are provided as S1 File with this article.

## Expected results

Using the protocols described, we performed experiments to validate the methods presented in this work.

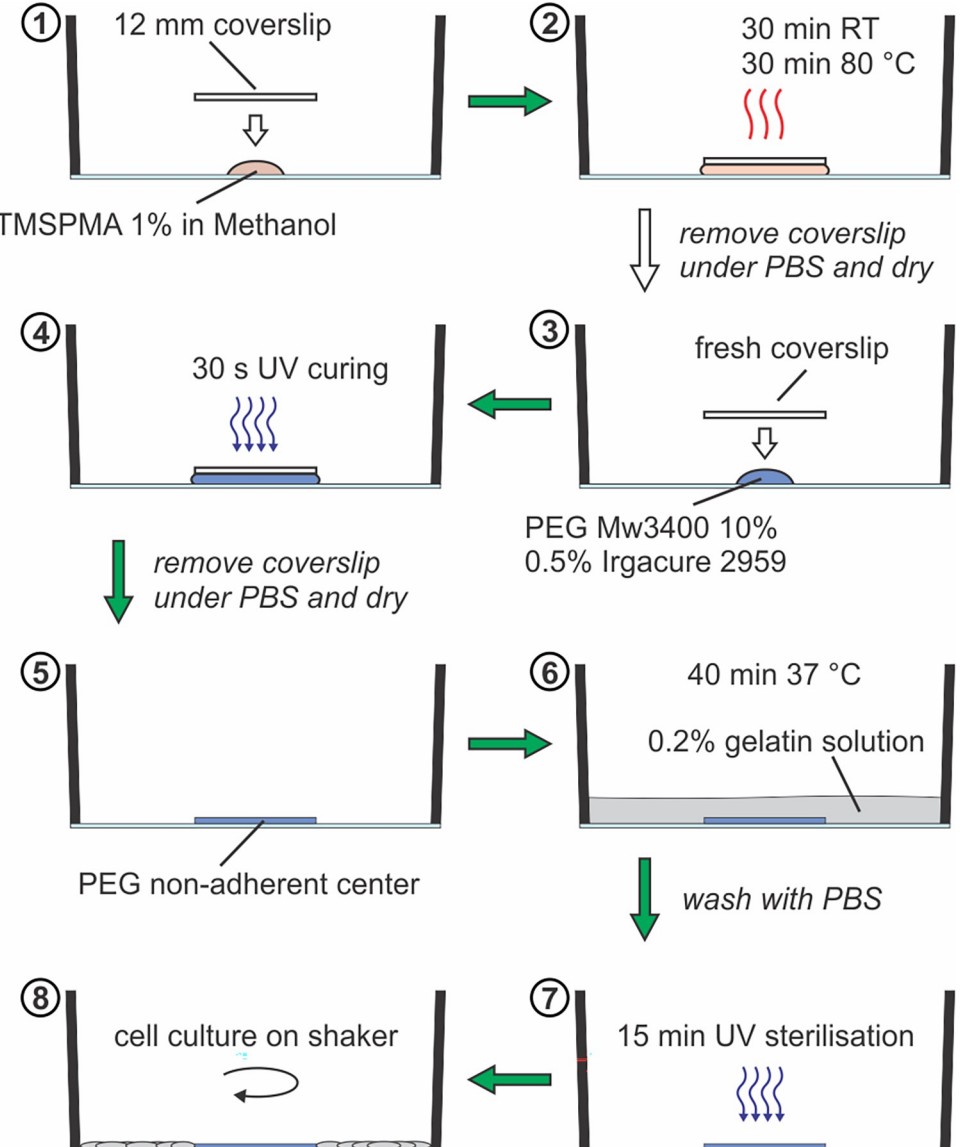

**Fig 2. Step-by-step description of the method to block cell adhesion in the center of the well using a PEG-hydrogel.** (1) Placing a drop of TMSPMA at the center of the well. (2) Covering the droplet with a glass coverslip for incubation. (3,4) Incubation with PEG Mw3400 and Irgacure 2959 using a fresh coverslip and curing through UV exposure. (5,6) Coating of the well with a 0.2% gelatin solution and washing. (7) Sterilization with UV light. (8) Cell culturing on an orbital shaker.

## PEG-gel coating prevents cell adhesion to regions of disturbed flow

To prevent cell growth in the region of disturbed flow, a patch of non-adhesive PEG-gel was applied to the center of a 6-well glass-bottom plate, as illustrated in Fig 2. As a model system, a co-culture of SMCs and ECs was grown on the orbital shaker for three days. SMCs were seeded first and stained with cell tracker after reaching confluency, and afterwards ECs were seeded on top. Co-cultures were exposed to shear stress on the orbital shaker for three days, fixed and ECs were immunostained with an anti-VE-cadherin antibody. A large mosaic image was created by stitching multiple single images, reconstructing half of the well. This allowed to confirm the complete absence of cells in the central region where PEG was present (Fig 3).

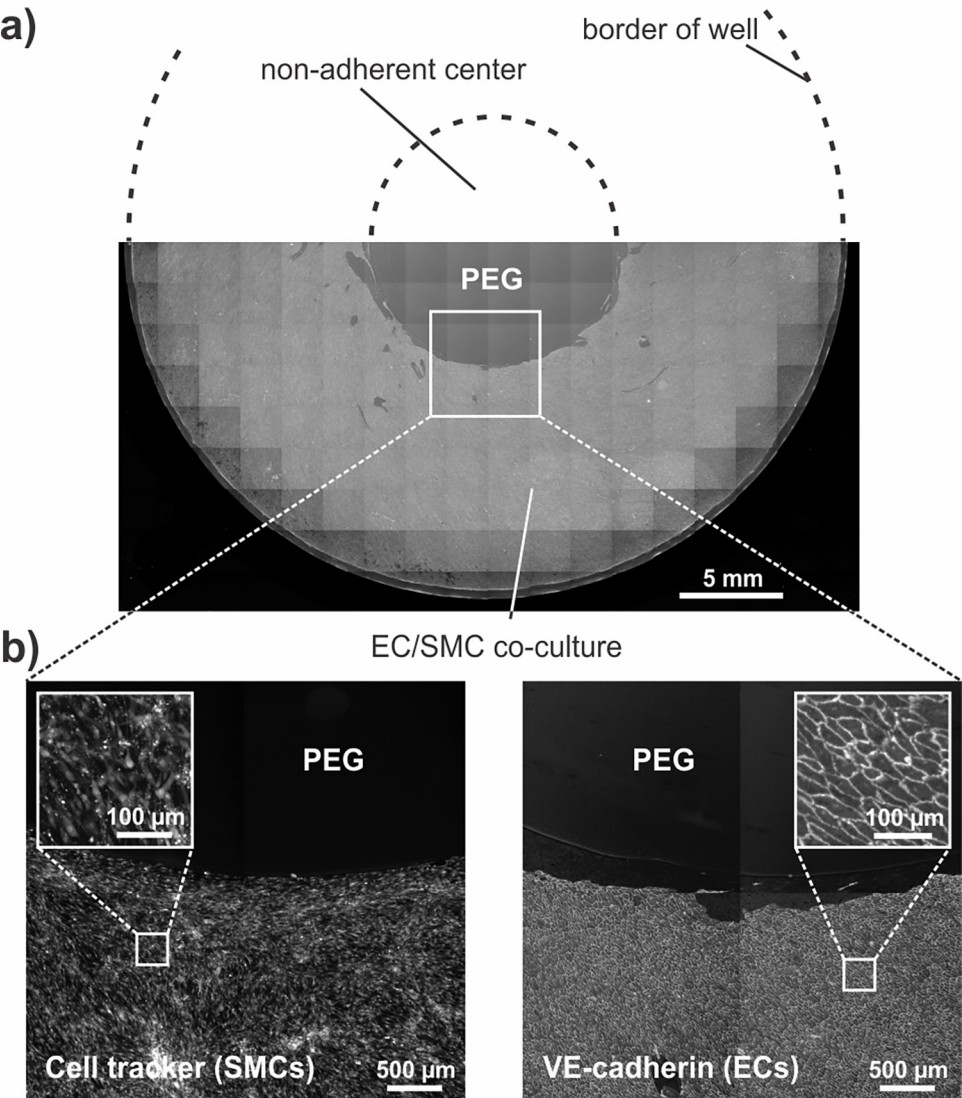

**Fig 3. PEG-gel patch preventing cell adhesion in the center of the well.** SMC-EC co-culture in a 6-well plate with a patch of PEG-gel in the center. **a)** The top half of the well is shown as a schematic, while the bottom half shows ECs stained with VE-cadherin immunostaining. Widefield fluorescent images were stitched into a large mosaic. Scale bar 5 mm. **b)** Zoomed-in images of the white square showing SMCs stained with cell tracker (left) and ECs with VE-cadherin immunostaining (right). Scale bar 500 μm.

Although the whole well was stained in this case, the multi-staining platform insert presented in the next section, could in theory also have been added before the staining. As previously shown in Ghim et al., segmentation had no effect on cell shape or alignment compared to culturing cells on the whole well [21].

## The multi-staining platform allows for multiple stainings on the same tissue

An EC monolayer was exposed to shear stress for three days on the orbital shaker. To achieve a flow profile like the one described by computer simulations in Salek et al. [18], the shaker was set to 135 rpm and 2 mL of culture medium were used per well. After cell fixation, the multi-staining platform was assembled on top of the monolayer, as shown in Fig 4.

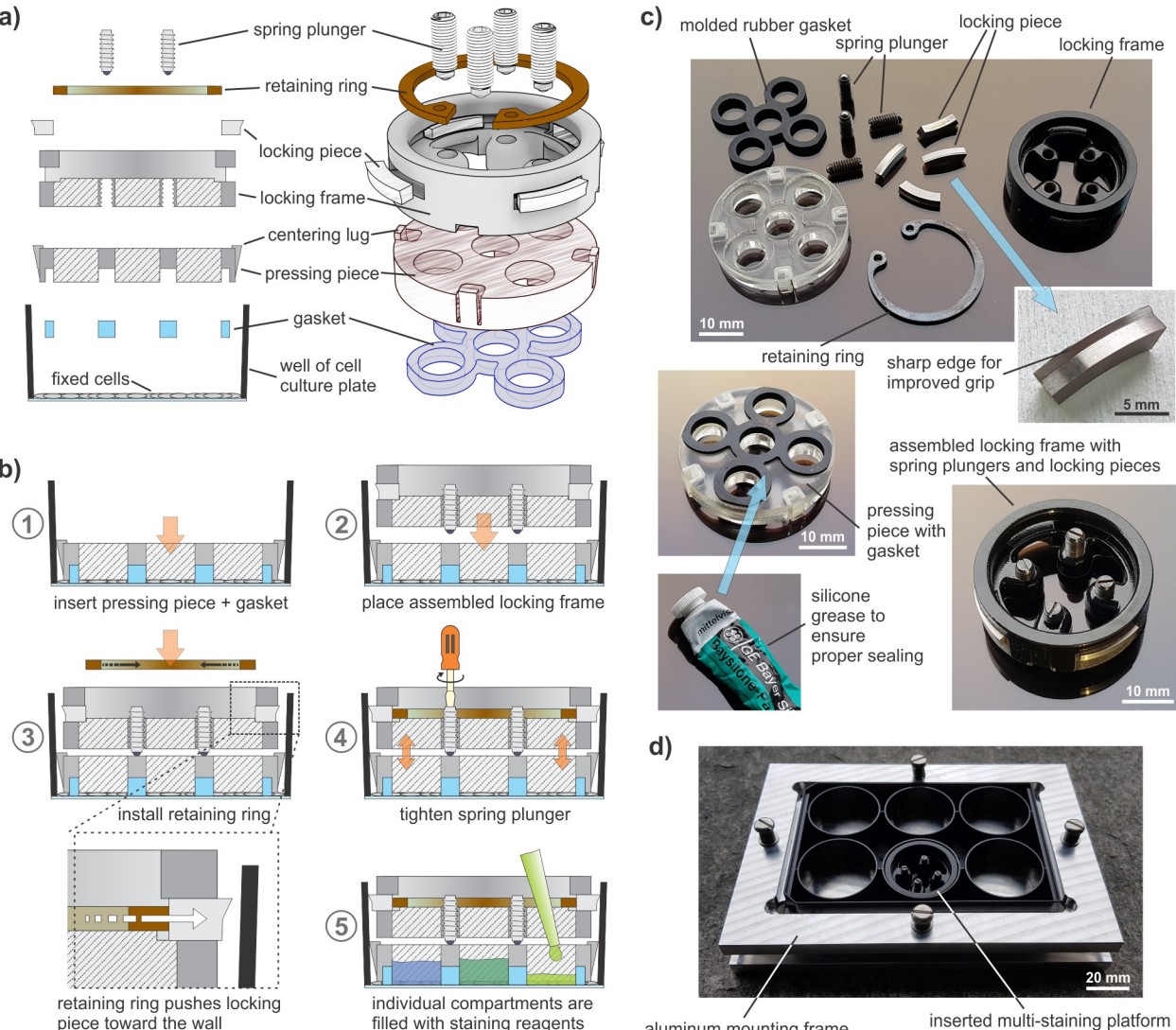

**Fig 4. Assembly of a multi-staining platform insert. a)** Overview schematics depicting the different pieces of the staining platform and assembly order. **b)** Assembly steps: (1) The pressing piece with the gasket is inserted into the well and placed directly on top of the fixed cells; (2) the locking frame is placed on top, and the centering lugs ensure proper alignment. The four locking pieces were previously inserted into the locking frame with the sharp edge facing upwards and the spring plungers inserted into the holes of the locking frame; (3) using circlip pliers, the retaining ring is compressed to fit into the groove of the locking frame, pressing the locking pieces out towards the wall of the well; (4) the spring plungers are tightened to apply vertical force; (5) the individual compartments are filled with staining reagents. **c)** Photographs of the different pieces and materials used for the assembly of a multi-staining platform insert. **d)** Photograph of the aluminum mounting frame, containing a glass-bottom 6-well plate with an inserted multi-staining platform.

As a proof-of-concept, VE-cadherin immunostaining and DAPI staining were performed in the central compartment and in one of the peripheral compartments. Fig 5 shows the successful isolation of the pockets and the high quality of the staining results. As expected for ECs exposed to shear stress on an orbital shaker, the alignment of the cells in the direction of the flow could be observed in the peripheral pocket, while the cells in the center were randomly oriented [9, 10].

A direct comparison between the center and the periphery using only 100 μL of staining solution per pocket was possible with our platform. Typically, 900 μL are necessary to evenly stain the whole well. The use of the platform therefore reduces the amount of required reagents

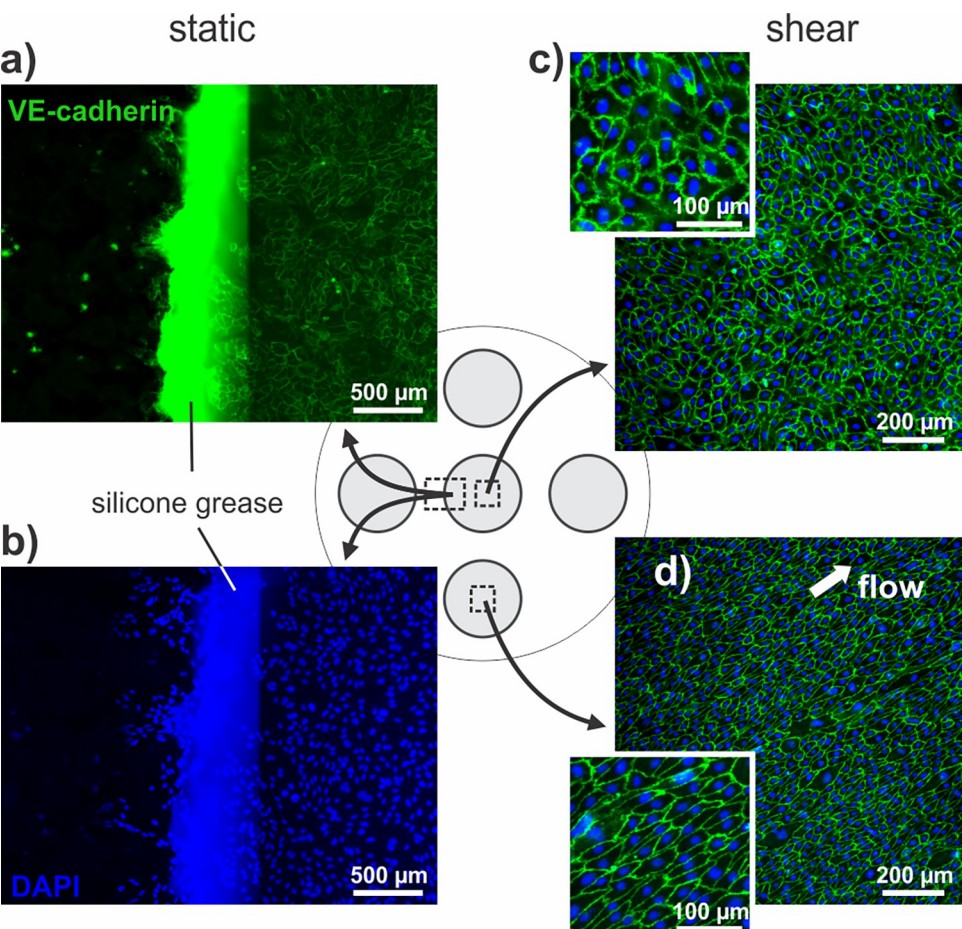

**Fig 5. VE-cadherin and DAPI stainings performed with the multi-staining platform.** A confluent EC layer was exposed to shear on an orbital shaker with 2 mL fluid and 135 rpm rotation speed, for three days. Images were acquired using a widefield fluorescent light microscope. VE-cadherin immunostaining (green) shows the plasma membrane of HUVECs. DAPI staining (blue) shows cell nuclei. **a)** and **b)** show the border of one pocket and silicone grease that is used for sealing. **c)** Shows an image of cells in the central compartment, which are randomly aligned because they have been exposed to disturbed flow. **d)** Shows an image of cells in the bottom peripheral compartment, which are aligned along the flow direction.

by almost 90%. After the staining was completed, 100 μL were removed from each pocket, confirming that there was no leakage.

The required drawings with instructions for the 3D-printing of the pieces, for both plastic-bottom and glass-bottom 6-well plates are provided as part of the protocol published on protocols.io.

## Discussion

Orbital shakers in combination with 6-well plates have great potential as experimental platforms to expose endothelial cells to shear for research in vascular biology. However, the complexity of the fluid mechanics and the limited versatility of the system may deter biologists from utilizing it, despite its ease of use, reliability, and availability in many labs. The advantages and disadvantages of orbital shakers are summarized in Table 1, along with the solutions presented in this work.

**Table 1. Advantages and disadvantages of orbital shakers and how we addressed some of the associated challenges.**

**ADVANTAGES**

- Relatively cheap to set up
- Possibility to study effects of different shear magnitudes and fluid flow profiles (center vs periphery)
- Potential to study large tissue areas without missing the "big picture" and possibly interesting observations
- High-content screening possible
- Mimics pulsatility in blood vessels
- Allows for collection of supernatants for secretome analysis

| DISADVANTAGES | PROPOSED SOLUTIONS |
| --- | --- |
| Heterogeneous shear and fluid flow profile create tissue heterogeneity (center vs periphery) that may affect cell phenotype and/or dilute and mask biological differences between sample groups. | Non-adherent center with PEG-hydrogel coating, to grow cells only at the periphery of the wells. Decreases sample heterogeneity and facilitates analysis and sample collection exclusively from the region exposed to physiological shear and flow profile. |
| Area is too large to perform immunostainings economically and to screen different markers. | Multi-staining platform to section the tissue into 5 smaller compartments, allowing to compare the center and the periphery and screen multiple stainings simultaneously. Increases throughput while reducing costs. |
| Relatively expensive in terms of biological material and reagent consumption during cell culture, compared to smaller multi-well plates. | Reduced overall costs of experiments through reduction of the required staining reagent volumes when using the staining platform. |
| Conventional plastic-bottom wells require long working distance microscope objectives due to their thickness, which hinders high-resolution imaging. | Use glass-bottom 6-well plates. The aluminum mounting frame will prevent breaking of the glass when a vertical force is applied through the staining platform. |
| Limited range of shear possible due to spilling danger with increased rotation speed and fixed well radius. | |
| Does not mimic other blood flow characteristics such as wall tension, viscosity, vascular resistance, and blood pressure. | |

The first limitation that we address is the tissue heterogeneity caused by different shear and fluid flow profiles in the center versus periphery of the well, which could mask biological differences between sample groups. Furthermore, knowing that ECs express very different phenotypes when exposed to different shear profiles, and that all secreted factors will be mixed in the culture medium, certain parameters could be influenced, as was also proposed by Ghim et al. [21]. Using non-adherent PEG gels as a cell-patterning method is a simple way of excluding cells from the regions of disturbed flow and achieving a more homogeneous cell population, for biological analyses, such as proteomics, RNA sequencing, Western Blot and RT-PCR. Moreover, due to the exclusion of the atheroprone central regions, supernatants can be collected to selectively analyze the secretome of the cells exposed to laminar pulsatile flow. We have shown that, in a co-culture with sequential seeding of SMCs and ECs, both cell types adhered exclusively to the periphery of the well (Fig 3). Although this method is time-consuming and requires precise manual handling, it does not interfere with the flow, nor does it require engineered devices, only materials and reagents that are readily available in a lab, contrarily to other published solutions [21, 22]. It will be a valuable quick and easy tool for studies focusing only on the effects of physiological shear, or for control experiments when it is unclear whether cells in the central region are affecting or masking the parameters that are being analyzed.

Theoretically, one could also mask the central part and coat the rest of the dish with PEG to study the effect of disturbed flow on the cells in the center of the dish, but instead of using coverslips a more complex masking strategy will be necessary, such as the one presented in Ghim

et al. [21]. Another simple alternative could be to place a soft PDMS circular block in the center, coating the periphery of the well with PEG and removing the PDMS block afterwards, a method that has not been tested in this study.

Another limitation of this system is that the wells on a 6-well plate are relatively large, and therefore relatively large amounts of biological sample, antibodies and other expensive reagents would be required when performing screenings and in-depth characterizations involving immunostaining and imaging. To avoid having to change to a different system, or putting anything in the well that could lead to contamination or interfere with the flow profile as characterized in the literature [18], we propose a multi-staining platform that compartmentalizes the well after cell fixation, thereby reducing the staining area. This platform allows to perform multiple parallel stainings and to compare central and peripheral regions within the same sample to investigate the effects of atheroprone versus atheroprotective flow profiles. Furthermore, because the platform is added only at the end of the experiment, the whole well can be analyzed and the supernatants collected before compartmentalizing for the staining, without missing potential interesting results when looking at the "whole picture". As a proof-of-concept, VE-cadherin and DAPI stainings were performed, confirming that handling did not interfere with the quality of the staining and the integrity of the tissue (Fig 5).

Although it seems trivial in theory, the compartmentalization of a tissue surface without leakage or tissue drought was quite challenging. Compartmentalization of a hydrophobic surface, such as the bare surface of a cell culture dish, is much simpler. In this case, the molded silicone gasket with silicone grease alone (Fig 4), a donut-shaped gasket made of soft PDMS, or a simple circle drawn with a DAKO pen could have done the trick. The challenge here was to properly isolate the compartments on the hydrophilic surface of a cellular tissue, without compromising its integrity, drying it, or interfering with the staining quality. Thus, a certain complexity was necessary to create appropriate vertical force without breaking the glass-bottom of the well. The usage of this platform is based on the fabrication of custom-made parts, which requires access to machines for milling and rapid prototyping. Once established, the assembly steps are relatively simple, and the pieces can be washed and reused.

We hope that our solutions will help and motivate scientists to explore the effects of shear on both cell monocultures and co-cultures using orbital shakers. The developed methods can also be adapted to fields other than vascular biology. This could benefit scientists in biomedical research in various endeavors, such as understanding the role of shear in cellular mechanisms in health and disease, finding new therapies, characterizing tissue engineered constructs or cell sheets, or studying co-cultures.

## Supporting information

**S1 File. Lab protocol from protocols.io with technical drawings and materials/methods combined as a PDF document.**
(PDF)

## Acknowledgments

The authors would like to thank Dr. Sara Motta and Prof. Simon Hoerstrup for isolating and providing HUVEC cells, Julia Mehl for the first tests with the multi-well inserts in our lab, Prof. Simone Schürle-Finke for supporting the microscopic imaging, and Roger Meier from the ScopeM facility for support with imaging and stitching large mosaics with JavaScript. From IBM Research we want to thank Dr. Emmanuel Delamarche and Dr. Heike Riel for their continuous support.

## Author Contributions

**Conceptualization:** Andreia Fernandes, Viola Vogel, Robert D. Lovchik.

**Data curation:** Andreia Fernandes.

**Funding acquisition:** Viola Vogel.

**Methodology:** Andreia Fernandes, Vahid Hosseini, Robert D. Lovchik.

**Resources:** Viola Vogel, Robert D. Lovchik.

**Visualization:** Robert D. Lovchik.

**Writing – original draft:** Andreia Fernandes, Robert D. Lovchik.

**Writing – review & editing:** Andreia Fernandes, Vahid Hosseini, Viola Vogel, Robert D. Lovchik.

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
