## [Decision Letter · Decision Letter 0]

17 Nov 2021

PONE-D-21-23497Engineering solutions for biological studies of flow-exposed endothelial cells on orbital shakersPLOS ONE

Dear Dr. Lovchik,

Thank you for submitting your manuscript to PLOS ONE. After careful consideration, we feel that it has merit but does not fully meet PLOS ONE’s publication criteria as it currently stands. Therefore, we invite you to submit a revised version of the manuscript that addresses the points raised during the review process.

We look forward to receiving your revised manuscript.

Kind regards,

Panayiotis Maghsoudlou

Academic Editor

PLOS ONE

Journal Requirements:

2. Thank you for providing the following Protocols.io DOI in the Methods section of your manuscript [Protocols.io DOI]. In keeping with our submission requirements, please add the Protocols.io DOI in the “Protocol DOI” field of the submission form (via “Edit Submission”). For more information, please see our submission guidelines: https://journals.plos.org/plosone/s/submission-guidelines#loc-guidelines-for-specific-study-types

"" 

5. Please amend the manuscript submission data (via Edit Submission) to include author Andreia Fernandes, Vahid Hosseini and Viola Vogel.

Reviewers' comments:

Reviewer's Responses to Questions

**Comments to the Author**

1. Does the manuscript report a protocol which is of utility to the research community and adds value to the published literature?

Reviewer #1: Yes

Reviewer #2: Yes

2. Has the protocol been described in sufficient detail?

Descriptions of methods and reagents contained in the step-by-step protocol should be reported in sufficient detail for another researcher to reproduce all experiments and analyses. The protocol should describe the appropriate controls, sample sizes and replication needed to ensure that the data are robust and reproducible.

Reviewer #1: Yes

Reviewer #2: Yes

3. Does the protocol describe a validated method?

Reviewer #1: Yes

Reviewer #2: No

4. If the manuscript contains new data, have the authors made this data fully available?

Reviewer #1: Yes

Reviewer #2: No

**5. Is the article presented in an intelligible fashion and written in standard English?**

Reviewer #1: Yes

Reviewer #2: Yes

6. Review Comments to the Author

Reviewer #1: The manuscript is clear, well-structured and contains the required protocol information in the supplementary materials file.

Reviewer #2: The paper “Engineering solutions for biological studies of flow-exposed endothelial cells

on orbital shakers” by Fernandes et al., describes two technical/engineering improvements on methods/devices for studying (endothelial) cells exposed to diverse flow patterns in an orbital shaker platform using circular cell culture chambers (wells of a 6-well plate).

General criticism:

1. The idea to avoid interference from heterogenous flow patterns (disturbed, oscillatory flow /recirculating eddies in the center vs, unidirectional, laminar flow) which elicit distinct physiological responses, is good, but not innovative. This idea has been promoted in an elegant 2018 paper by Ghim et al. The relatively small engineering innovation here is in the specific way, how the central zone of the culture dish is rendered non-adherent. A critical question, however, that has not been addressed here, is whether this really matters, i.e., whether signals originating from the cells in the disturbed central zone impact the cells elsewhere in the culture dish? The Ghim paper is quite ambivalent about this (“depends on what parameter you are looking at”). Comparisons of the physiological responses of their cells (e.g. expression / upregulation of EC activation molecules) in the laminar flow zones in the presence or the absence of cells in the center of circular culture dishes, would increase the confidence in this methodology .

2. The proposed approach for generating a multi-staining platform is technically elegant, but, in principle, not novel: A similar idea has been published by Ghim et al., 2018. This new paper seems to build on and extend the Ghim paper. Moreover, the engineering approaches for creating this multi-staining platform, appear to be technically involved and fairly cumbersome. Previous work by others (e.g., Brooks et al., Endothelium, 11:45–57, 2004) has employed a much simpler, yet effective approach, placing small circular coverslips in various locals of a circular culture dish.

3. The authors should validate their system by computational fluid dynamics (CFD) since their experimental conditions are similar to, but no identical to those used by Ghim et al.

Specific comments

1. The authors’ description of the flow patterns in the center of the dish as “turbulent’ (3 occurrences in the text) is incorrect. Rather, it is characterized as disturbed/oscillatory. For a more quantitative, engineering approach, that authors should characterize their system by CFD.

2. Immunostaining should not just be limited to visualizing VE-cadherin for EC identification, but should be extended to includee functional analysis assessing the expression/upregulation of activation molecules, such as VCAM-1 or E-Selectin

3. Figure 3 b is not publication-quality, the distinct staining patterns (immune) are not discernible.

4. Figure 4 is instructive, but in its complexity indicative for the highly complicated engineering approach. Much simpler solutions have been described in the past (see, e.g., Brooks et al., 2004)

5. Figure 5 , Panels a and b are not publication-quality

7. PLOS authors have the option to publish the peer review history of their article (what does this mean?). If published, this will include your full peer review and any attached files.

Reviewer #1: **Yes: **Patricia Diaz-Rodriguez

Reviewer #2: **Yes: **Peter I Lelkes

---

## [Author Response · Author response to Decision Letter 0]

14 Dec 2021

please see attached response letter

---

## [Editor Report · Decision Letter 1]

16 Dec 2021

Engineering solutions for biological studies of flow-exposed endothelial cells on orbital shakers

PONE-D-21-23497R1

Dear Dr. Lovchik,

We’re pleased to inform you that your manuscript has been judged scientifically suitable for publication and will be formally accepted for publication once it meets all outstanding technical requirements.

Kind regards,

Panayiotis Maghsoudlou

Academic Editor

PLOS ONE

---

## [Editor Report · Acceptance letter]

11 Jan 2022

PONE-D-21-23497R1 

Engineering solutions for biological studies of flow-exposed endothelial cells on orbital shakers 

Dear Dr. Lovchik:

I'm pleased to inform you that your manuscript has been deemed suitable for publication in PLOS ONE. Congratulations! Your manuscript is now with our production department. 

Kind regards, 

on behalf of

Dr. Panayiotis Maghsoudlou 

Academic Editor

PLOS ONE